# "We pray for the night to be shorter, so we can change our menstrual pads": A qualitative exploration of menstrual hygiene challenges among internally displaced adolescent girls in Northern Ethiopia, 2023

**Balem Demtsu Betsu**[1]*, **Araya Abrha Medhanyie**[2], **Tesfay Gebregzabher Gebrehiwet**[2], **L. Lewis Wall**[3,4,5]

1 Department of Midwifery, College of Health Sciences, Mekelle University, Mekelle, Ethiopia, 2 School of Public Health, College of Health Sciences, Mekelle University, Mekelle, Ethiopia, 3 Department of Anthropology, College of Arts and Sciences, Washington University in St. Louis, St. Louis, MO, United States of America, 4 Department of Obstetrics & Gynecology, Washington University in St. Louis, St. Louis, MO, United States of America, 5 Department of Obstetrics & Gynecology, Ayder Comprehensive Specialized Hospital, College of Health Sciences, Mekelle University, Mekelle, Ethiopia

☺ These authors contributed equally to this work.

* balemdim@gmail.com

## Abstract

### Background

Menstrual hygiene management is a critical aspect of adolescent health. However, access to adequate menstrual hygiene products and sanitation facilities is severely restricted during times of war. There is a dearth of information about the menstrual hygiene needs of adolescent girls during humanitarian crises. This study investigated the menstrual hygiene management needs of the internally displaced adolescent girls in the war-torn region of Tigray, Ethiopia.

### Methods

A qualitative study was conducted in three centers for internally displaced people in Mekelle, Tigray Region. Four focus-group discussions and six in-depth interviews were conducted using the local language among 39 adolescent girls aged 13–19 years. The collected data were recorded, transcribed, translated, and analyzed using the ATLAS.ti-7.5.18 software through a qualitative thematic analysis approach.

### Results

Six primary themes were identified: 1) shortage of menstrual pads; 2) poor accommodation of latrine facilities; 3) silence around menstruation; 4) exchange of menstrual pads for life-saving commodities, 5) lack of privacy; 6) menstruation is a "curse" for adolescent girls living in displaced people's camps. The study highlights the significant challenges faced by internally displaced adolescent girls in managing their menstrual hygiene while living in camps

**Data Availability Statement:** All relevant data are within the manuscript and its Supporting Information files.

**Funding:** The author(s) received no specific funding for this work.

**Competing interests:** LLW serves as a non-compensated member of the board of directors of the charity, Dignity Period. The other authors have no competing interests to declare. This does not alter our adherence to PLOS ONE policies on sharing data and materials.

for internally displaced people. Lack of access to adequate menstrual hygiene management supplies often leads them to use rags or worn-out clothing instead of menstrual pads.

## Conclusion

Access to adequate menstrual hygiene products and sanitation facilities is severely limited among displaced adolescent girls in Tigray. The findings emphasize the urgent need to address menstrual hygiene issues during emergencies. Supplying essential items such as sanitary pads, soap, water, sanitation infrastructure, and improved information on menstrual hygiene management should be prioritized.

## Introduction

Internally displaced persons, known as IDPs, are individuals who have been compelled to leave their homes due to conflict, violence, persecution, or disasters, yet they stay within their country's borders. Worldwide, 68.3 million internally displaced persons comprise the largest segment of the forcibly displaced population at 58 percent. Internally displaced persons are among the most vulnerable people globally [1].

Ethiopia is one of the five sub-Saharan countries reporting the highest figures of internally displaced persons at the end of 2023, according to the Global Report on Internal Displacement 2023 [2]. The internal displacement was caused by conflict and violence, and by flooding. According to the OCHA Ethiopia Situation Report 2024, 4.4 million internally displaced people are severely impacted by conflict, hostilities, and climate shocks, where Tigray (27.4 percent) accounts for the highest internally displaced people caseloads nationwide [3].

War and other humanitarian crises that lead to the displacement of populations and overcrowding in internally displaced people camps disproportionately affect girls and women due to the critical shortage of water, sanitation, and hygiene facilities in temporary camps and shelters [4, 5]. Inadequate sanitation puts the safety and dignity of adolescent girls at risk, increasing their vulnerability to social and economic challenges, negative health outcomes, gender-based violence, and human rights issues, during humanitarian crises [4, 6, 7].

Menstrual hygiene management is an integral part of adolescent health and well-being. During humanitarian crises and natural disasters the proportion of displaced women and girls accounts for at least 50% [8]. Adolescent girls are often disproportionately affected by the crises of inadequate menstrual hygiene management, which leads to significant physical and psychological health risks. They often encounter multiple difficulties in these situations, including a higher risk of violence and exploitation, restricted access to education and healthcare, family separation, diminished social support networks, and discrimination based on their gender and ethnicity [9, 10]. Displaced Adolescent girls are highly challenged due to severely limited access to menstrual hygiene products and adequate sanitation facilities [11]. The integration of menstrual hygiene management into humanitarian response highlights three key components of successful programs: providing reliable information and education concerning menstruation, ensuring access to basic materials and menstrual supplies, and creating a supportive infrastructure for those affected [12–14]. In camps for internally displaced people, the provision of reliable menstrual information and the development of a supportive infrastructure are often overlooked.

To improve the health and well-being of adolescent girls during times of war, it is crucial to address the gaps in their specific menstrual hygiene management needs and experiences.

Although efforts have been made to address menstrual hygiene management during humanitarian emergencies, there is still much to learn about the challenges faced by adolescent girls in war-torn regions. Studies conducted among adolescent girls in their home communities in northern Ethiopia have shown that they often experience embarrassment, fear of harassment, and shortages of menstrual hygiene management materials and supplies. This is coupled with community misperceptions and misinformation concerning menstruation [15, 16]. However, the menstrual experiences of adolescent girls who have been forced to flee from their homes into internal displacement camps are not adequately explored. This is the first study that attempted to assess menstrual hygiene management experiences among adolescent girls displaced due to the war eruption on November 4[th], 2020, in the Tigray Region.

## Materials and methods

### Study design and setting

A qualitative study using phenomenological design was undertaken in 2023 to explore the menstrual experiences of adolescent girls staying in camps for internally displaced people in Tigray. Tigray region hosed for around 949,371 internally displaced people residing on 667 sites. The highest internally displaced people concentrations are in North Western zone (240,728), Mekelle (215,289), and Central zone (187,732). Conflict caused 99% of displacements, with most IDPs living with host communities (80%) [17].

The study took place in Mekelle town, the capital of the Tigray region that hosts the second-highest number of internally displaced persons in the region. Mekelle is home to 222, 310 internally displaced people centers and a large population of displaced people [17]. The study was conducted in three camps for internally displaced people within the city. Following the conflict from 2020 to 2022, Mekelle has experienced a significant influx of displaced people fleeing from various parts of the region.

### Sampling procedure and sample size

A purposive sampling technique was implemented to recruit study participants. The camp coordinators assigned to each center for internally displaced people were informed about the purpose and procedure of the study. The camp coordinators contacted the mobile health nutrition teams in the camp lead to identify adolescent girls as study participants for in-depth interviews and focus-group discussions.

### Study population and eligibility criteria

Internally displaced adolescent girls who reached menarche during or before the displacement to the camp participated in the study.

### Data collection tool and procedure

We conducted the interviews using open-ended questions, followed by further probing after each question to expand the insight provided regarding their experiences of menstrual hygiene management in these camps. A series of six individual in-depth interviews were conducted from December 10, 2022, to January 20, 2023, to explore further aspects of the situation in detail in addition to the focus-group discussions. At the start of each interview, the research project was briefly explained to the participants. We strove to put the participants at ease and to assure them of the confidentiality of their responses. Both the moderator and the audio recorder were female, to help put the participants at ease. The focus group discussions and interviews were conducted by the principal author, who is a female and a native speaker of

Tigrigna (the local language), assisted by a female research assistant with qualitative research expertise.

## Data analysis

We collected the data using an audio recording, while field notes were also taken. The interviews were transcribed, translated into English, and imported into ATLAS.ti Version 7.5.18, Scientific Software Development Mnbh, Berlin, for analysis. After reading and re-reading the data several codes were identified that attributed to the development sub-themes. Codes were organized into families resulting in six major themes and six subthemes. Thematic data analysis was carried out concurrently with data collection to ensure that the information collected was sufficient to answer the research questions. This helped to identify gaps in the data and the level of saturation and inform subsequent data collection efforts.

## Ethical consideration

Ethical clearance for the study was obtained from the institutional review board of the College of Health Sciences, Mekelle University (reference number MU-IRB1980/2022. We obtained an official letter of support introducing the project from Mekelle University, the School of Public Health. Parents of adolescent girls who were under 18 years old during the data collection period gave their written consent for their daughters to participate in the research. The participants themselves also provided their written assent after being informed of the purpose and objective of the research. They were told that they could choose not to answer any questions that made them uncomfortable and that their participation was completely voluntary. The participants were also assured that their personal information would remain confidential.

We conducted the interviews using the local language (Tigrigna) inside the tents that were arranged by the coordinators to provide privacy and confidentiality. No other people were present or within hearing distance during the interviews. To maintain confidentiality, unique codes were assigned to each participant, ensuring that no names or personal information were disclosed.

## Results

A total of 39 adolescent girls aged 13–19 years residing in three internally displaced centers of Mekelle for more than a year were interviewed (Table 1). Six primary themes and six

**Table 1. Socio-demographic characteristics of study participants.**

| S.no | Variables | Frequency | Mean age |
|------|-----------|-----------|----------|
| 1. | **Age** | | |
| | 13–15 year-old | 22 | 15.72 ± 1.43 years old |
| | 16–19 year-old | 17 | |
| 2. | **Age at menarche** | | |
| | 13–18 year-old | 39 | 13.77±1.22 years old |
| 3. | **School Grade** | | |
| | 4–8 grade | 35 | |
| | 9–10 grade | 4 | |
| 4. | **Ethnicity** | | |
| | Tegaru | 39 | |
| 5. | **Religion** | | |
| | Orthodox Christian | 31 | |
| | Muslim | 8 | |

subthemes were identified from the analysis of the data. 1) Shortage of menstrual pads; 2) poor accommodation of latrine facilities; 3) silence around menstruation; 4) exchange of menstrual pads for life-saving commodities; 5) lack of privacy; 6) menstruation is a "curse" for adolescent girls living in displaced people's camps.

## Shortage of menstrual pads

The adolescent girls emphasized that the shortage of menstrual pads poses major challenges for meeting their menstrual hygiene needs in the camps. The participants agreed that purchasing sanitary pads under the conditions imposed by being forced to live in a camp was simply not possible. They could not afford to buy menstrual supplies in their current circumstances. They all agreed that before the war broke out, they had better access to menstrual supplies, soap, and water, as well as private spaces for managing menstruation. As one of the focus-group participants explained:

*Currently, I am not using menstrual pads. But when I was at home, I used to buy every type of menstrual pad. But now I have nothing [money] to buy with. I could not even ask my parents for money to buy pads Emm. . .not only menstrual pads, but they also [parents] don't even have money to buy food.* (18-year-old, focus group discussant, December 2022)

Another interviewee expressed her thoughts by saying:

*It is quite different from what we were doing at home [menstrual hygiene management]. This is the worst condition we have ever had in our life. When our menstruation comes, it is scary, especially when we know we do not have a menstrual pad! We do not get soap! We do not get underwear! We need pads to change at intervals and soap to wash our pads, but we have none! We are so worried here [in the camp!].* (15 years old, in-depth interviewee, December 2022)

The lack of access to menstrual pads, soap, and underwear has left them feeling scared and worried during their menstruation. The absence of necessities for menstrual hygiene management has made their situation the worst they have ever experienced. Participants reported using worn-out cloths as an alternative to menstrual pads. These were the most frequently mentioned materials that the adolescent girls used for menstrual hygiene, followed by washable/reusable pads because other options were not affordable.

*Emm. . .when it [menstruation] happens, while we were at home, I mean. . .before we were displaced, we used modess [disposable pads] but now we cannot get modess and we are using pieces of clothing.* (16-year-old, focus group discussant, January 2023). (Modess is the brand name of a particular menstrual pad that has been picked up in Ethiopia as a general name for the product)

Although ragged cloths and washable pads are the options that displaced adolescent girls must absorb their menstrual flow accessing worn-out cloths is another challenge.

## Wearing clothes or using them for menstrual absorption

Because they were often forced to leave their homes quickly and were unable to gather their belonging before fleeing, study participants had limited clothing with them they resorted to cutting up the few clothes they had with them and using them for menstrual hygiene purposes.

This shortage of clothes makes it difficult for them to maintain proper menstrual hygiene. A fifteen-year-old focus group discussant stated the challenge this way:

*We can't find old cloths easily [to tear and use to make pads]. While we were at home, we used to find as many clothes as we needed. However, here [in the camp] we don't even have enough clothes to change we cannot tear up clothes for that purpose [menstrual absorbent] as there is a critical shortage of clothes. We suffer a lot because of menstruation, and we always curse the day we were born.* (15-year-old focus group discussant, December 2022)

Another discussant also highlighted the challenge as,

*It is very difficult to tear our clothes for menstrual hygiene. If we plan to use our clothes as a menstrual pad, we will have a shortage of clothes and we cannot tear our clothes for this purpose, but we are using our pajamas or whatever we have.* (16-year-old Girl, December 2022)

The participants noted that not all types of clothing are suitable for menstrual hygiene purposes. The soft, absorbent fabric was identified as the most comfortable and effective to manage menstrual flow. Adolescent girls must be on the lookout for old clothes that meet these requirements, even if they have to borrow them from others. Pajamas were mentioned as a preferable type of cloth to use for menstrual hygiene purposes. Other types of fabric may be difficult to wash, making blood stains difficult to remove even with the girl's best efforts. Failure to remove bloodstains effectively may also lead to shame and stigma if such stains are visible.

### Inadequate water and soap to wash the menstrual pads

Due to the shortage of menstrual absorbent materials, adolescent girls often reuse them during their next menstrual cycle. The study participants often mentioned, however, that re-using menstrual absorbents comes up with its additional challenge which is a lack of soap and water. The camps for internally displaced people do not meet the need for water and soap to clean menstrual absorbents. This creates additional challenges for maintaining menstrual hygiene and sanitation in the camps.

### Poor accommodation of latrine facilities

Displaced adolescent girls often struggle to maintain menstrual hygiene due to the limited availability of adequate sanitation facilities. The latrines in the camps are used by a large number of people, are located far away from their living accommodations, have poor hygiene, and most importantly do not have lockable doors. As a result, adolescent girls face the following challenges with respect to using latrines.

### The latrines are too busy/overcrowded

The participant worried about using the latrine to change their menstrual pads because of the long waiting time. They also spend more time struggling to properly change and wash their menstrual absorbents. One girl expressed her feelings about spending too much time in the latrine for menstrual hygiene purposes this way:

*Using the toilet is inconvenient as many people are using it, and if we decide to use it [toilet] we feel ashamed and are forced to leave almost immediately. This is because if we stay inside the toilet for a longer time, people will suspect that we are doing something (such as changing*

*a menstrual pad) and they will try to see inside through the door hole. . .* (17- year-old, in-depth interviewee, January 2023)

### Lack of separation between males and females

Even though latrines are limited in number, they are not segregated by sex they are all used by males and females together. This makes using the latrine extremely uncomfortable for adolescent girls. Another concern that the participants emphasized is that the shared latrines are often unclean, unsafe, and smell bad, making it difficult for girls to spend the time necessary to remove, wash, and replace their menstrual pads while in the latrine.

### The latrines are located too far

From their tents to make using them safe or convenient at night. Most girls expressed their concern that visiting the latrine at night exposes them to potential sexual and gender-based violence, as well as the possibility of being attacked by wild animals such as hyenas. If they are not accompanied by someone else when they go to the latrine at night, they worry about endangering their lives.

*Let me share with you what happened to me one time. One night, as usual, there was no light in the camp, and I had to go to the toilet. I had to light a candle to get there. Unfortunately, the candle was blown out by the wind, and I got scared and had to change the modess [menstrual pad] in the dark. (*15-year-old, in-depth interviewee, December 2022)

Another participant expressed her concern, saying:

*When menstruation happens at night, we suffer, because there is no light here [in the camp]. We cannot go to the toilet because it is too dark, and the toilet is far away. There are also hyenas and dogs wandering around during the night, so we just suffer and worry. We can't do anything until it is dawn, but we pray for the night to be shorter so we can change our menstrual pads.* (16-year-old focus group discussant, December 2022)

The experiences shared by the internally displaced adolescent girls in the camp highlight the significant challenges they faced while dealing with menstruation in a setting with inadequate lighting and safety concerns at night.

### The absence of lighting in camps

For internally displacement people poses significant challenges for adolescent girls during menstruation. This underscores the unmet need for sanitation facilities and lighting to uphold the basic dignity and safety of adolescent girls during menstruation.

### Silence around menstruation

The cultural shroud of secrecy surrounding menstruation presents a challenge for adolescent girls in displacement camps. In the camps for the internally displaced people, there are cramped living arrangements, where families and even strangers share tents which further worsens the silence due to the lack of privacy. This lack of privacy hinders open discussions about reproductive health, making it difficult for adolescent girls to access information, especially for those who need accurate information and support regarding menstrual hygiene management, which is often considered a taboo topic in their societies.

*Oh, that (discussing menstruation) is unthinkable. Even back at home discussing menstruation is not comfortable except with my mom. When we come here, people's concern is about safety, food, and returning home. So, we don't discuss it (menstruation). Can't you see how many of us are in a single tent? How can we ask for information in front of all the family members here?* (13 years old, in-depth interviewee, January 2023)

Another discussant said:

*We don't have information about the availability of health services . . .. I don't know whether there is a health center here inside the camp. For instance, the idea of going to a clinic to get pain medication during menstruation has never come to my mind because I don't even know if there is a clinic here (in the camp).* (14 years old, focused group discussant, January 2023)

The absence of open discussion leaves girls unprepared to effectively manage their menstrual hygiene, with potentially harmful effects on their health and well-being. In the context of displacement camps, this cultural silence becomes an even heavier burden, hindering access to information and support networks that could empower adolescent girls to navigate their periods with dignity and confidence.

## Exchange of menstrual pads for life-saving commodities

Access to food and medicines is severely limited among the internally displaced population. To overcome these challenges, adolescent girls are forced to exchange the menstrual pads they receive for food and other priorities. One of the focus-group discussants explained:

*We are rarely provided with menstrual pads and soap, and not all girls and women in the camp receive them. So, whatever we get, we exchange it for food and fall back on using worn-out clothes as menstrual pads* (15-year-old focus group discussant, January 2023)

Furthermore, in a critical moment when a family member became seriously ill and essential medication was unavailable at a local health facility, the girls had to sell their allocated menstrual pads to buy the medication from a private vendor. This act of sacrifice was necessary to save the life of one of the study participants, as elaborated below, underscoring the harsh realities experienced by the displaced adolescent girls

*Once, my youngest brother felt severely ill with diarrhea and vomiting, but the government health center nearby did not have the necessary medication, so we had to purchase it from a private drug seller. Unfortunately, we (the family) did not have enough money to buy the medication, but something sprang into my mind: I could sell the menstrual pads that I had received. I had to raise the amount of money we needed. It was difficult, but it was necessary to save my brother's life. I am sure I would not die from using old clothes for menstrual pads, but my brother could have died. . .* (18-year-old In-depth interviewee, January 2023)

## Lack of privacy

The community's unfavorable attitudes towards menstruation, lack of adequate personal space in the living areas, lack of access to clean toilets, and no private places to wash and dry reusable menstrual pads all constantly threaten the privacy of the displaced girls. Privacy threats often

come from family members (especially boys), as well as from other displaced people living in the camp. One adolescent girl stated her concern saying.

*Oh, it is very difficult! When menstruation comes at night we cannot go to the toilet because it is so far away. We cannot change the pads inside our home because it is dark inside and if we try to search for a pad at home during the night everybody will be awakened and asked, 'What is going on with you?' So, we prefer to keep silent and wait until morning* (16-year-old, In-interviewee, December 2022)

If menstrual products are to be reused for the next cycle, they must be washed, dried, and stored properly, but the physical set-up of the camps makes it difficult to do this. Normally many family members share a single tent with people whom they didn't know before. There is no place to hang menstrual cloths for drying, which makes it difficult to keep them out of sight. Hanging them out either inside or outside the tent allows everyone to see them from all directions. Sometimes the girls wash their menstrual pads and cover them up with cloths, which causes them to stay wet, and become spoiled and odiferous. One informant explained:

*. . . We do not have the freedom to do as we wish, so we get worried. We do not hang the pads outside freely as we do with other clothes. As my friends mentioned before, we throw away reusable pads and clothes, which could have been reused if we could have washed and dried them* (17-year-old, focus group discussant, December 2022)

Another participant said:

*. . . So, the only chance we get to wash our menstrual pads is during the night, but the problem is how and where can you hang it up to dry?. . .* (18-year-old, Focus group discussion January 2023)

This indicates the inability to hang pads outside freely and the challenge of finding a suitable place to wash and dry menstrual pads during the day or night. The inability to reuse washable pads and clothes due to the lack of proper washing and drying facilities further adds a challenge for adolescent girls in internally displaced camps.

## Menstruation is a "curse" for adolescent girls living in displaced people's camps

A strong opinion held by many participants is that they considered menstruation as a "curse" rather than a blessing when living in the camps. The lack of access to menstrual hygiene management materials, their economic cost, and negative community perception of menstruation accompanied by the traumatic experiences engendered by being forced from their homes by conflict, led many girls to consider menstruation as a "curse" or "punishment from the Almighty. Quotes like these capture the girl's views:

*I came on foot to this camp, crossing up and down the country. While I was traveling, my period appeared, and I really cursed it because I had nothing to use to absorb it. Before this time [before war and displacement], I was eager to see it every month. However, when menstruation appeared while I was traveling [during displacement], I hated myself for being a female. . .oh. . . [Sigh of desperation].* (18-year-old, in-depth interviewee, January 2023)

Another interviewee shared her perception saying:

*You know what I wish every time my menstruation occurs here in the camp. . .? I wish I had never been born, or that I had been born male! . . . Anyway. . . we have no option except to accept nature.* (16 years old, in-depth interviewee, January 2023)

The expression of frustration and self-loathing at the lack of menstrual hygiene materials during the journey to the camp has drastically changed the perception of menstruation from a natural occurrence to a source of grief of distress. Adolescent girls would have preferred to have been born male, not to face the menstrual-related challenges. Accordingly, the lack of access to menstrual hygiene resources has a profound impact on the mental and emotional well-being of adolescent girls in displacement camps.

## Discussion

This study highlights the exacerbated challenges of menstrual hygiene faced by adolescent girls living in internally displaced peoples' camps due to the humanitarian emergency in Tigray. Shortage of menstrual pads; poor accommodation of latrine facilities; silence around menstruation; exchange of menstrual pads for life-saving commodities; lack of privacy; and perceiving menstruation as a "curse" while living in the internal displacement camp are the common challenges encountered.

People who are displaced due to conflict, environmental issues, and socioeconomic factors face many challenges, such as a lack of menstrual hygiene materials, limited access to water and sanitation, privacy concerns, and few options for managing menstrual materials [5, 18, 19]. In such emergencies, moreover, the security, privacy, and health needs of women and girls including menstrual health are often overlooked [18, 20, 21] This study depicts the criticality of lack of menstrual hygiene materials and supplies which is in line with other studies [9, 22–24]. Due to their circumstances, girls and women have few options regarding menstrual management supplies and are often forced to use unclean, unsatisfactory materials, and damp underwear to manage their periods [6, 22]. Similar other studies in low and middle-income countries showed that menstruators face challenges in obtaining water and soap for washing materials and garments, as well as finding discreet areas for drying menstrual absorbents [25, 26].

In our study, we found that the lack of privacy for changing, washing, drying, storing, and disposing of menstrual products was a significant issue. The camp shelters, (tents) and the latrines are not women and girls friendly. The latrines were inaccessible at night due to the absence of lighting and lockable doors, and the toilets were not segregated by gender which is in line with findings from other studies [27–29].

While the Interagency Agency Standing Committee (ISAC) 2019 and Sphere Handbook advocate for adequate lighting in camp settings, and UNFPA dignity kits include solar-powered torch lights to prevent violence and insecurity [30–32], some adolescent girls sell the kits, including the torch lights, to alleviate financial difficulties. Consequently, they are apprehensive about using the latrine at night due to the fear of being attacked by assailants or wild animals in the darkness. This aligns with the findings of a study in Uganda, where displaced girls expressed fear of gender-based violence when accessing water, hygiene, and sanitation facilities [33].

Another major theme that emerged during the analysis in this study is how the girls' perceptions of menstruation changed after the war and displacement. Before becoming a refugee, adolescent girls used to think of menstruation as a "gift from God" to women that allowed them to have children. However, after the conflict drove them from their homes they now think of menstruation as a "curse". Adolescent and young women refugees in Brazil similarly

describe menstruation using negative terms like "horrible," "terrible," "bad," or "painful" [34]. The multifaceted challenges faced by adolescent girls during displacement have led to this change in perception.

Sommer et al. Stressed the importance of providing accurate information about managing menstruation in a way that bolstered girls' confidence and allowed them to manage their periods with confidence and without fear or shame [35]. Access to reliable menstrual information and improved menstrual knowledge have a positive impact on menstrual hygiene management and lessen negative psychosocial impacts [4, 36].

Providing instructions on how to use the supplies in humanitarian settings is a crucial component of the success of menstrual interventions in such settings [1]. Nevertheless, in many humanitarian situations, the distribution of kits without instructions has resulted in beneficiaries encountering difficulties in using them [18]. A study in Vanuatu described that receiving unfamiliar menstrual hygiene management kits caused confusion about how to use the products [37]. The inadequate and irregular distribution of sanitary pads combined with limited menstrual information and education posed significant challenges like menstrual anxiety and stigma for adolescent girls [4, 16, 18, 38]. This may be exacerbated among minority groups who find themselves among the refugee population [14].

It's important to understand that menstrual hygiene management is not just about cleanliness and personal hygiene, but also a critical aspect of adolescent girls' overall health. Without access to accurate information and resources, girls may face health complications such as infections and skin irritation. The lack of open discussion and limited access to information about reproductive health can lead to shame and stigma, impacting mental health and self-esteem [36]. Therefore, creating safe spaces for open discussion and access to resources is crucial for effective menstrual hygiene management.

This study has some limitations. Due to security concerns and restricted mobility during the time of data collection, we collected the data exclusively from camps located in Mekelle. This may have restricted the information we could gather from other study participants. Additionally, the data collection process could have been strengthened by triangulating it with observation to provide a more comprehensive understanding of the collected data. However, the study's strengths lie in its use of triangulated data collection methods using focus group discussions and in-depth interviews. Besides, to the best of the authors' knowledge, this is the first study of its kind in the study area. Therefore, it will inform stakeholders in identifying priorities for implementation.

## Implications for future research and programs

The study findings have significant implications for researchers, policymakers, and practitioners in the field of humanitarian settings and menstrual hygiene management. This finding can help to highlight the urgent need for further research on improving menstrual hygiene management in war-affected areas. This includes investigating the impact of menstrual hygiene management supplies, facilities, and education on the well-being of adolescent girls. By conducting more research in this area, researchers can contribute to the development of effective interventions and policies to address the challenges faced by displaced adolescent girls.

Practitioners, on the other hand, should prioritize the provision of menstrual hygiene management supplies with clear instructions and focus on creating women and girls-friendly facilities with adequate privacy and security measures. This includes ensuring that the distribution of sanitary pads is regular and accompanied by clear instructions for use.

For policymakers, the study underscores the need to address critical issues related to menstrual hygiene management in war-torn areas. Policymakers should implement measures to

ensure the regular distribution of sanitary pads and improve facilities to meet the unique needs of women and girls in such environments. This may involve developing policies that prioritize the creation of safe spaces, access to resources, and improved facilities to support effective menstrual hygiene management.

## Conclusion

The results of this study highlight the significant challenge that internally displaced adolescent girls encounter when trying to maintain their menstrual hygiene. These girls find themselves in a difficult situation due to their restricted access to menstruation pads, lack of trustworthy information, and inadequate water, sanitation, and hygiene facilities. The lack of menstrual absorbent material, the usage of soiled cloths in place of menstruation pads, the scarcity of water and soap for washing, and the girls' restricted access to latrines have all had a substantial negative influence on their capacity to properly manage their period hygiene. Moreover, among the girls who have been displaced, the stigma attached to menstruation has been exacerbated by the absence of privacy in terms of both physical space and social attitudes. These factors collectively highlight the urgent need for comprehensive interventions to address the multifaceted challenges faced by these vulnerable adolescent girls.

## Supporting information

**S1 Data.**
(DOC)

## Author Contributions

**Conceptualization:** Balem Demtsu Betsu, Araya Abrha Medhanyie, Tesfay Gebregzabher Gebrehiwet, L. Lewis Wall.

**Data curation:** Balem Demtsu Betsu.

**Formal analysis:** Balem Demtsu Betsu.

**Investigation:** Balem Demtsu Betsu.

**Methodology:** Balem Demtsu Betsu, Araya Abrha Medhanyie, Tesfay Gebregzabher Gebrehiwet, L. Lewis Wall.

**Supervision:** Balem Demtsu Betsu, Araya Abrha Medhanyie, Tesfay Gebregzabher Gebrehiwet, L. Lewis Wall.

**Validation:** Balem Demtsu Betsu, Araya Abrha Medhanyie, Tesfay Gebregzabher Gebrehiwet, L. Lewis Wall.

**Writing – original draft:** Balem Demtsu Betsu.

**Writing – review & editing:** Balem Demtsu Betsu, Araya Abrha Medhanyie, Tesfay Gebregzabher Gebrehiwet, L. Lewis Wall.

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
