## [Decision Letter · Decision Letter 0]

3 Jul 2024

PONE-D-24-12948Date: March 31/2024 To: Plos One

“We pray for the night to be shorter, so we can change our menstrual pads”: A qualitative exploration of menstrual hygiene challenges among internally displaced adolescent girls in Northern Ethiopia, 2023PLOS ONE

Dear Dr. Betsu,

Thank you for submitting your manuscript to PLOS ONE. After careful consideration, we feel that it has merit but does not fully meet PLOS ONE’s publication criteria as it currently stands. Therefore, we invite you to submit a revised version of the manuscript that addresses the points raised during the review process.

Dear authors,

You are advised to make changes before reevaluation of the manuscript.

with regards,

Ranjit

We look forward to receiving your revised manuscript.

Kind regards,

Ranjit Kumar Dehury

Academic Editor

PLOS ONE

2. During your revisions, please note that a simple title correction is required: please remove the follwing "Date: March 31/2024 To: Plos One" from the title. Please ensure this is updated in the manuscript file and the online submission information.

“LLW serves as a non-compensated member of the board of directors of the charity, Dignity Period. The other authors have no competing interests to declare.”

Additional Editor Comments:

Dear authors,

You are advised to make changes before reevaluation of the manuscript.

with regards,

Ranjit

Reviewers' comments:

Reviewer's Responses to Questions

**Comments to the Author**

1. Is the manuscript technically sound, and do the data support the conclusions?

Reviewer #1: Yes

Reviewer #2: Partly

2. Has the statistical analysis been performed appropriately and rigorously? 

Reviewer #1: Yes

Reviewer #2: No

3. Have the authors made all data underlying the findings in their manuscript fully available?

Reviewer #1: Yes

Reviewer #2: Yes

4. Is the manuscript presented in an intelligible fashion and written in standard English?

Reviewer #1: Yes

Reviewer #2: Yes

5. Review Comments to the Author

Reviewer #1: This is an extremely pertinent research topic, especially in developing countries like Ethiopia where access to WASH facilities is limited. The study area, the Tigray region, is heavily impacted by this issue because of the ongoing warfare in the area. Moreover, the paper was well written. However, I have some concerns regarding the introduction, methods, discussion, and conclusion sections.

1. Your introduction section lacks adequate documentation please add more researches which is relevant to your study.

2. Organizing the methods section in a structured way is preferable.

3. Please make the appropriate corrections as you failed to provide a detailed description of your study area in the methods section.

4. Please revise the discussion to take into account the following points: highlight any intriguing or unexpected results and link them to the research question, mention the study's limitation, and suggest ways in which it might be applied to increase the body of knowledge in your area.

5. Please make recommendations for future research directions in your conclusion section.

Reviewer #2: More literature needs to be incorporated, and brief information about the environment of displacement camps needs to be provided.

What are the specific regions where the menstrual pad should be changed? Explain with existing evidence from the literature.

As the Authors adopted the phenomenological design in this study, the study's theoretical implication adds more value to the paper.

The authors should explain the study area briefly. What are the characteristics of study setups? Explain all the factors that influence adolescent girl's behaviour towards menstrual hygiene.

Authors should explain the themes and sub-themes in the methodology parts. What is the process followed for the selection of themes?

Briefly explain the data saturation methods for both in-depth interviews and FGD.

The authors should explain respondents' social and demographic factors before presenting the study results. Also, explain the characteristics of the current place of residence.

Is it due to a shortage of products or a lack of purchasing power of the participant?

Quotes the narrative about social stigma and embarrassment they faced in society. Also, how long did the participants reside in the camps?

Author can use the pseudonym of participate. So the reader can connect the narratives with the participant.

6. PLOS authors have the option to publish the peer review history of their article (what does this mean?). If published, this will include your full peer review and any attached files.

Reviewer #1: **Yes: **Tarikuwa Natnael

Reviewer #2: No

---

## [Author Response · Author response to Decision Letter 0]

29 Jul 2024

Dear reviewers it is an honor to have your inputs for the betterment of the manuscript. The comments and questions have critical role to make our paper readable. Hence we have made the requested clarifications, modifications and descriptions as per the reviewers recommendation. 

Reviewer #1: Yes

Reviewer #2: Partly

Author response: Dear review we have implemented qualitative study and we have detailed the methodology in a way it can ensure replicability. We did not do experiment and control in this case. And the sample size is determined by data saturation as detailed in the method section 

2. Has the statistical analysis been performed appropriately and rigorously?

Reviewer #1: Yes

Reviewer #2: No

Author response:

As mentioned above and indicated in line # 82 the methodology is qualitative study using phenomenological design for this reason there is no need to do statistical analysis

3. Have the authors made all data underlying the findings in their manuscript fully available?

Reviewer #1: Yes

Reviewer #2: Yes

4. Is the manuscript presented in an intelligible fashion and written in standard English?

Reviewer #1: Yes

Reviewer #2: Yes

5. Review Comments to the Author

Reviewer #1: This is an extremely pertinent research topic, especially in developing countries like Ethiopia where access to WASH facilities is limited. The study area, the Tigray region, is heavily impacted by this issue because of the ongoing warfare in the area. Moreover, the paper was well written. However, I have some concerns regarding the introduction, methods, discussion, and conclusion sections.

Reviewer Review Comments to the Author Author Response

Reviewer #1 Your introduction section lacks adequate documentation please add more researches which is relevant to your study We have incorporated additional literatures to elaborate the introduction section as per the reviewer recommendation ( line # 48-58)

 Organizing the methods section in a structured way is preferable. We have structured the method section in a structured way as per the recommended (line # 92-149.)

 Please make the appropriate corrections as you failed to provide a detailed description of your study area in the methods section. On the current version of the manuscript we have made detailed description the study setting , as per your valued comment (line # 94-106)

 Please revise the discussion to take into account the following points:

Highlight any intriguing or unexpected results and link them to the research question, mention the study's limitation, and suggest ways in which it might be applied to increase the body of knowledge in your area.

 • Most of the study findings are in line with other studies of menstrual hygiene management challenges in humanitarian settings. One unexpected result in this study is the change in perception about menstruation. As it is indicated in line # 393-400, though the adolescent girls perceive menstruation as “Gift from God” the displacement made them to perceive it differently. 

• Study limitation is depicted in line 422-424

 Please make recommendations for future research directions in your conclusion section. The recommendations are included in the “Implication for further research and program” section (Line# 431-448)

Reviewer #2 More literature needs to be incorporated, and brief information about the environment of displacement camps needs to be provided. On the current version of the manuscript we have made detailed description the study setting , as per your valued comment (line # 94-106)

 What are the specific regions where the menstrual pad should be changed? Explain with existing evidence from the literature.

 As indicated in line # 383-387 other literatures have shown that place for changing pad is critical challenge and there is no specific place (region) for changing pads in displacement camps, which is in line with our study. 

“In our study, we found that the lack of privacy for changing, washing, drying, storing, and disposing of menstrual products was a significant issue. The camp shelters, (tents) and the latrines are not women and girls friendly. The latrines were inaccessible at night due to the absence of lighting and lockable doors, and the toilets were not segregated by gender which is in line with findings from other studies” 

 As the Authors adopted the phenomenological design in this study, the study's theoretical implication adds more value to the paper. The authors should explain the study area briefly. What are the characteristics of study setups? Explain all the factors that influence adolescent girl's behavior towards menstrual hygiene. 

 Study area is described in the current version 

Factors that hindered management of menstrual hygiene in the camps have come out as major themes in the result section; which include:

Shortage of menstrual pads( line # 163), Poor accommodation of latrine facilities ( line #227), Silence around menstruation (line 271), and Lack of privacy (line #295)

 Authors should explain the themes and sub-themes in the methodology parts. The down listed themes and sub themes are indicated in the result section and the reason why we did not include it in the method section is to avoid repetition of ideas. 

Six major themes( Line # 154-158)

Six subthemes 

1. Wearing clothes or using them for menstrual absorption; (line # 195)

2. Inadequate water and soap to wash the menstrual pads( Line #220)

3. The latrines are too busy/overcrowded ( line # 233)

4. Lack of separation between males and females (line # 242)

5. The latrines are located too far (248); 

6. The absence of (267) lighting in camps)

 What is the process followed for the selection of themes? Briefly explain the data saturation methods for both in-depth interviews and FGD The procedures followed to identify the themes are clarified in this version in line # 131-136. 

“After reading and re-reading the data several codes were identified which attributed to the development sub-themes. Codes were organized into families resulting in six major themes and six subthemes. Thematic data analysis was carried out concurrently with data collection to ensure that the information collected was sufficient to answer the research questions. This helped to identify gaps in the data and the level of saturation and inform subsequent data collection efforts.” 

Moreover, data saturation was determined to be present when replications took place and no new codes or themes emerged during the preliminary analysis

 The authors should explain respondents' social and demographic factors before presenting the study results. Also, explain the characteristics of the current place of residence. 

 Social and demographic characteristics are indicated in this version Table-1 line #159-161 

The study setting is detailed in the updated version of the manuscripts in line # 94-106

 Is it due to a shortage of products or a lack of purchasing power of the participant? 

 As indicated in the major themes it is both shortage of products and lack of economic power that hindered the adolescent girls from accessing the menstrual pads 

 The Quotes the narrative about social stigma and embarrassment they faced in society.?

 The issue of social stigma is highlighted in line number 271 

“Silence around menstruation”

And the Quotes are is in line #279 and # 285

 Also, how long did the participants reside in the camps The participants have resided in the camps for more than a year (Line # 153-154)

“A total of 38 adolescent girls aged 13-19 years residing in three internally displaced centers of Mekelle for more than a year were interviewed”

---

## [Decision Letter · Decision Letter 1]

22 Aug 2024

Date: March 31/2024 To: Plos One

“We pray for the night to be shorter, so we can change our menstrual pads”: A qualitative exploration of menstrual hygiene challenges among internally displaced adolescent girls in Northern Ethiopia, 2023

PONE-D-24-12948R1

Dear Dr. Betsu,

We’re pleased to inform you that your manuscript has been judged scientifically suitable for publication and will be formally accepted for publication once it meets all outstanding technical requirements.

Kind regards,

Ranjit Kumar Dehury

Academic Editor

PLOS ONE

Additional Editor Comments (optional):

Dear authors,

After taking into evaluation the comments of the reviewers the paper is accepted.

With regards,

Ranjit

Reviewers' comments:

Reviewer's Responses to Questions

**Comments to the Author**

1. If the authors have adequately addressed your comments raised in a previous round of review and you feel that this manuscript is now acceptable for publication, you may indicate that here to bypass the “Comments to the Author” section, enter your conflict of interest statement in the “Confidential to Editor” section, and submit your "Accept" recommendation.

Reviewer #1: All comments have been addressed

Reviewer #2: All comments have been addressed

2. Is the manuscript technically sound, and do the data support the conclusions?

Reviewer #1: Yes

Reviewer #2: Yes

3. Has the statistical analysis been performed appropriately and rigorously? 

Reviewer #1: Yes

Reviewer #2: Yes

4. Have the authors made all data underlying the findings in their manuscript fully available?

Reviewer #1: Yes

Reviewer #2: Yes

5. Is the manuscript presented in an intelligible fashion and written in standard English?

Reviewer #1: Yes

Reviewer #2: Yes

6. Review Comments to the Author

Reviewer #1: Dear authors,

I appreciate that you responded to all of the comments. Before publishing, could you also proofread the manuscript for grammar mistakes?

Reviewer #2: Author needs to read it very carefully and arrange the narratives in a scientific flow.

7. PLOS authors have the option to publish the peer review history of their article (what does this mean?). If published, this will include your full peer review and any attached files.

Reviewer #1: **Yes: **Tarikuwa Natnael

Reviewer #2: No

---

## [Editor Report · Acceptance letter]

28 Aug 2024

PONE-D-24-12948R1 

PLOS ONE

Dear Dr. Betsu, 

I'm pleased to inform you that your manuscript has been deemed suitable for publication in PLOS ONE. Congratulations! Your manuscript is now being handed over to our production team.

Kind regards, 

on behalf of

Dr. Ranjit Kumar Dehury 

Academic Editor

PLOS ONE